# Transcription Factor CmNAC34 Regulated *CmLCYB*-Mediated β-Carotene Accumulation during Oriental Melon Fruit Ripening

**DOI:** 10.3390/ijms23179805

**Published:** 2022-08-29

**Authors:** Yaping Zhao, Xiaoyu Duan, Lixia Wang, Ge Gao, Chuanqiang Xu, Hongyan Qi

**Affiliations:** Key Laboratory of Protected Horticulture of Education Ministry and Liaoning Province, College of Horticulture, Shenyang Agricultural University, National and Local Joint Engineering Research Centre of Northern Horticultural, Facilities Design and Application Technology (Liaoning), Shenyang 110866, China

**Keywords:** β-carotene, *CmLCYB*, CmNAC34, transcription regulation, oriental melon (*Cucumis melo*)

## Abstract

Ripened oriental melon (*Cucumis melo*) with orange-colored flesh is rich in β-carotene. Lycopene β-cyclase (LCYB) is the synthetic enzyme that directly controls the massive accumulation of β-carotene. However, the regulatory mechanism underlying the *CmLCYB*-mediated β-carotene accumulation in oriental melon is fairly unknown. Here, we screened and identified a transcription factor, CmNAC34, by combining bioinformatics analysis and yeast one-hybrid screen with *CmLCYB* promoter. CmNAC34 was located in the nucleus and acted as a transcriptional activator. The expression profile of *CmNAC34* was consistent with that of *CmLCYB* during the fruit ripening. Additionally, the transient overexpression of CmNAC34 in oriental melon fruit promoted the expression of *CmLCYB* and enhanced β-carotene concentration, while transient silence of CmNAC34 in fruit was an opposite trend, which indicated CmNAC34 could modulate *CmLCYB*-mediated β-carotene biosynthesis in oriental melon. Finally, the yeast one-hybrid (Y1H), electrophoretic mobility shift assay (EMSA), β-glucuronidase (GUS) analysis assay, and luciferase reporter (LUC) assay indicated that CmNAC34 could bind to the promoter of *CmLCYB* and positively regulated the *CmLCYB* transcription level. These findings suggested that CmNAC34 acted as an activator to regulate β-carotene accumulation by directly binding the promoter of *CmLCYB*, which provides new insight into the regulatory mechanism of carotenoid metabolism during the development and ripening of oriental melon.

## 1. Introduction

Oriental melon (*Cucumis melo* var. *makuwa* Makino) is a vital, fleshy fruit consumed worldwide due to its excellent fruit quality, nutrient content, texture, and flesh color. Flesh color is one of the most important indices to evaluate fruit quality, and the common cultivars show green, white, and orange-fleshed colors [1,2]. The formation of orange-fleshed color is mainly contributed to carotenoid accumulation in oriental melon. In plants, carotenoid content and composition differences confer considerable variation in color to root, flower, and fruit, which helps attract pollinators and spread seeds [3,4]. In addition, carotenoids participate in photosynthesis by promoting green-blue light absorption and energy transfer, and they also play an essential role in photoprotection reactions [5]. Carotenoids act as precursors for abscisic acid (ABA) and strigolactones (SLs) biosynthesis, which are involved in many physiological activities and stress responses [6,7]. In view of human health, carotenoids are regarded as potent antioxidants for scavenging reactive oxygen species (ROS) and other free radicals, which significantly reduces the risk of chronic diseases that may result from the excessive production of ROS, such as cancer, cardiovascular diseases, and diabetes [8]. Moreover, carotenoids are a source of vitamin A, which has a critical role in vision and immune responses [9]. 

In past decades, the pathway of carotenoid biosynthesis has been clearly characterized in a variety of plant species [4]. Moreover, due to significant differences in carotenoid components and content, the different key genes that contribute to carotenoid biosynthesis were identified in various horticultural crops. Lycopene accumulation is caused by the increased expression of carotenoid synthesis genes, *PSY* and *PDS*, during tomato (*Solanum lycopersicum*) ripening [10]. The massive accumulation of capsanthin positively correlates with the expression *PSY-1* and *DXS* in chill pepper (*Capsicum annuum*) [11]. The accumulation of β-carotene and lutein is associated with the expression of *LCYB* during kiwifruit (*Actinidia deliciosa*) fruit ripening [12]. The primary carotenoid composition, β-carotene, is accumulated in the peel and flesh of red-flesh loquat (*Eriobotrya japonica*) during maturation, which is consistent with increased expression of *PSY*, *LCYB*, and *BCH* [13]. The total carotenoid accumulation in yellow-fleshed peach (*Prunus persica*) is partially linked with increased transcripts of *PpFPPS*, *PpGGPS*, *PpLCYB*, and *PpCHYB* [14]. β-carotene and (E/Z)-phytoene are dominant carotenoid metabolites, and their accumulation is related to the expression of *PSY*, *NCED1*, and *CCD4* in orange-flesh apricot (*Prunus armeniaca*) [15]. Therefore, carotenoid composition and content, along with carotenogenic genes playing essential roles in carotenoid metabolism, are diverse and precise in various plant species.

Carotenoid metabolism is regulated by multiple factors, including development cues, environmental signals, and the expression of carotenogenic genes [16]. Transcription factors (TFs) represent primary regulators that operate the transcription of genes involved in carotenoid metabolism, which modify the carotenoid content. Up to now, many kinds of TFs regulated carotenoid metabolism have been identified. Tomato (*Solanum lycopersicum*) is usually used as model species to investigate fruit quality, and its carotenoid metabolism regulation network has been widely studied. Phytochrome-interacting factor PIF1a binds to the promoter of the *SlPSY1* gene to repress its expression, which eventually inhibits carotenoid biosynthesis [17]. The RIPENING INHIBITOR (RIN), a member of the MADS-box family, positively regulates tomato fruit ripening and activates the expression of *SlPSY1* to stimulate carotenoid biosynthesis [18,19]. SlMYB72 regulates multiple genes, *SlPSY1*, *SlZ-ISO*, and *SlLCYB*, to influence carotenoid content [20]. Moreover, other TFs, such as several SlWRKYs protein, SlBBX20, also directly participate in carotenoid metabolism [21,22].

NAC (NAM, ATAF1/2, and CUC2) proteins are plant-specific transcription factor families and are widely involved in distinct biological processes, such as plant immunity, fruit ripening, phytohormone biosynthesis, and secondary metabolism. For example, several NAC proteins have been identified to respond to abiotic stress, such as salt, heat, drought, and oxidative stress in rice [23,24,25]. In addition, increased expression of *SlNAC1* and reduced expression of *SlNAC4* in tomato (*Solanum lycopersicum*) lead to delayed fruit ripening accompanied by decreased ethylene synthesis, reduced carotenoid biosynthesis, and suppressed chlorophyll breakdown [26,27]. CrNAC036 in citrus (*Citrus reticulata*) and PpNAC.A59 in peach (*Prunus persica*) influence abscisic acid (ABA) and ethylene biosynthesis by regulating the expression of key genes in these pathways, respectively [28,29]. In recent years, NAC transcription factors that participate in secondary metabolism have received more attention. For instance, both CpNAC1 and CpNAC2 positively mediate carotenoid biosynthesis by directly activating carotenoid biosynthesis-related genes during papaya (*Carica papaya*) fruit ripening [30,31]. Moreover, MdNAC42 interacts with MdMYB10 to regulate anthocyanin accumulation in red-flesh apple (*Malus domestica Borkh*) fruit [32].

The flesh color of oriental melon is a critical trait determined by carotenoid content and composition. β-carotene is the dominant carotenoid component, accounting for more than 50% of the total carotenoid in orange-flesh oriental melon reported by a previous study [33]. In the present study, CmNAC34 was identified by yeast one-hybrid screening with *CmLCYB* promoter, and its expression profile was consistent with that of *CmLCYB* during fruit ripening, and its transcript level was significantly higher in orange-fleshed oriental melon. Additionally, *Agrobacterium tumefaciens*-mediated fruit infiltration assay was conducted to further verify that CmNAC34 could regulate *CmLCYB-*mediated carotenoid accumulation. Finally, Y1H, EMSA, LUC, and GUS assays all indicated that the CmNAC34 could directly bind to the promoter of *CmLCYB* and promote its transcription. In this study, we aimed to understand the regulatory mechanisms underlying carotenoid metabolism in oriental melon and hope to provide a theoretical basis for improving fruit quality.

## 2. Results

### 2.1. Accumulation of β-Carotene Increased during Oriental Melon Fruit Ripening

According to target metabolome analysis in a previous study, β-carotene is a major carotenoid metabolite in oriental melon, making the flesh of the ‘HDB’ fruit appear orange [33]. The results measured by HPLC showed that the β-carotene content of fruit was less than 10 μg·g^−1^ FW at the young fruit stage (from 10 to 20 days after anthesis, DAA), and the β-carotene content increased sharply at the mature stage from 21 μg·g^−1^ FW to 110 μg·g^−1^ FW (from 25 to 35 DAA) (Figure 1a). The L* value (negative-positive) represented from darkness to brightness, a* value (negative-positive) indicated from green to red, and b* value (negative-positive) showed from blue to yellow. The L* value sharply declined, and a* value and b* rapidly increased at the mature fruit stage (Figure 1b). The variation of these values was consistent with that of β-carotene content (Figure 1c), and these data indicated that 25 DAA is an essential point of flesh color change.

### 2.2. Expression Analysis of Carotenoid Metabolism Related Genes

The expression profiles of carotenoid metabolism-related genes were analyzed using the RT-qPCR in the previous study [33]. The results showed that these genes were initially upregulated and subsequently downregulated, peaking at 25 DAA, which implies that carotenoid accumulation was attributed to the expression of carotenoid biosynthesis-related genes. *CmLCYB* is a major enzyme that directly synthesis β-carotene; therefore, it was considered the target gene for further study. Additionally, the expression profile of *CmLCYB* was determined using an RT-qPCR experiment in three cultivars with different flesh colors, ‘HDB’ (orange flesh), ‘YMR’ (white flesh), and ‘XSM’ (white flesh). The results showed that the expression of *CmLCYB* in ‘HDB’ was significantly higher than in ‘YMR’ and ‘XSM’, which further suggested that *CmLCYB* plays a vital role in carotenoid biosynthesis (Figure 2a).

### 2.3. Putative Cis-Elements and Activity Analysis in CmLCYB Promoter

The *CmLCYB* promoter region (1474 bp) was submitted to PlantCARE and PLACE online databases to predict the regulatory *cis*-elements (Table 1). The core sequence, TATA-boxes, and CAAT-boxes were identified in numerous positions within the *CmLCYB* promoter region, which suggested the strong function in initiating *CmLCYB*. Except for these basal *cis*-elements, the *CmLCYB* promoter region also contained a series of putative *cis*-acting elements that refer to phytohormone signal response, diverse stress response, light signal response, and specific TFs binding motif. For example, the ABRE motif (CGTACGTG), CGTCA-motif (CGTCA), GARE-motif (TCTGTTG), TGA-element (AACGAC), and ERE element (ATTTTAAA) were phytohormone regulation elements that respond to abscisic acid (ABA), methyl jasmonate (MeJA), gibberellin (GA), auxin and ethylene, respectively. Few cis-elements related to abiotic stress were found in the *CmLCYB* promoter region, such as ARE motif and MBS motif. These *cis*-elements like NAC-motif (ACATGTG), MADS-box (CCTAAA), MYB (TAACCA), MYC (CAATTG), and W-box (TTGACC) are binding sites for specific TFs, which implies that *CmLCYB* could be regulated by various TFs. These results suggest that the transcriptional activity of the *CmLCYB* promoter may be manipulated by diverse environmental signals and TFs.

Meanwhile, the promoter sequence was divided into different fragments, fused to the GUS gene, and expressed in tobacco leaves. Therefore, the promoter activity was estimated through GUS activity and histochemical staining (Figure 2b,c). The results showed that different deletions of promoters all possessed transcription activity, and the transcription activity was promoted as the promoter region increased in length, which was consistent with histochemical staining.

### 2.4. Expression Profile, Subcellular Localization, and Transcriptional Activity of CmNAC34

Previous studies revealed that NAC TFs could mediate fruit ripening and bind to NACBS (NAC-binding site) in the promoter of genes related to carotenoid metabolism to influence carotenoid content [27,31]. A transcription factor, CmNAC34, was identified through yeast one-hybrid screening with the *CmLCYB* promoter fragment. In order to investigate the spatial and temporal expression profile of *CmNAC34*, the transcript level of *CmNAC34* was detected in different tissues and different developmental stages of fruit ripening using RT-qPCR assay. The result showed that the expression level of *CmNAC34* was higher in leaves, flowers, and fruit tissues than in roots, stems, and tendrils. During fruit ripening, the expression of *CmNAC34* was gradually upregulated, peaked at 25 DAA, and then slightly downregulated, which is consistent with the expression profile of *CmLCYB* (Figure 3a).

Additionally, we examined the subcellular localization of CmNAC34 through transient expression of CmNAC34 in tobacco leaves. We observed that the green fluorescent protein (GFP) signal of CmNAC34 overlapped with the signal of the nuclear maker, DAPI, which indicated that the CmNAC34 protein was localized in the nucleus. These results provided direct evidence that *CmNAC34* encoded a nuclear protein and could serve as a TF regulate expression of its target gene (Figure 3b). To further confirm the transcriptional activity, the Y2H Gold yeast cell transformed with recombinant plasmid pGBKT7-CmNAC34 was cultured in SD/-Trp for 3–5 days and then transferred to SD/-Trp-His-Ade with X-α-gal medium. The result showed that all yeast cells grew well on SD/-Trp medium, but only positive control and pGBKT7-CmNAC34 yeast cells could survive and turn blue on SD/-Trp-His-Ade with X-α-gal medium, which proved the CmNAC34 possessed transcriptional activity (Figure 3c).

### 2.5. CmNAC34 Regulates CmLCYB-Mediated Carotenoid Metabolism

To further examine the involvement of CmNAC34 in the alteration of carotenoid metabolism, the *CmNAC34* was transiently expressed through *Agrobacterium tumefaciens* infiltration in oriental melon fruit. The infiltrated fruit was sampled on different days after injection for further study. As the infiltration time continued, the luciferase fluorescence intensity gradually increased and then slightly decreased (Figure 4a). Meanwhile, RT-qPCR analysis revealed that the expression of *CmNAC34* and *CmLCYB* in CmNAC34-OE fruit was significantly higher than that of LUC-0 fruit, while their expression in CmNAC34-AN fruit was significantly lower than that of PRI101-0 fruit. These results imply that the transient expression of *CmNAC34* was successful in oriental melon fruit (Figure 4b,c). In addition, the expression profile of carotenoid biosynthesis-related genes was analyzed using RT-qPCR, and the result showed that the transcription levels *CmPSY1*, *CmPDS*, and *CmZDS* were not significantly different between LUC-0 and CmNAC34-OE fruit. The same results were presented between PRI101-0 and CmNAC34-AN fruit (Appendix A). Then, we detected the β-carotene content of transgenic fruits using HPLC analysis. Compared with the corresponding control, β-carotene content of CmNAC34-OE fruit was significantly increased at 2.5, 4.5, and 6.5 days after injection and significantly reduced in CmNAC34-AN fruit at 4.5 and 6.5 days after injection, respectively (Figure 4f,g). Overall, these findings suggested that CmNAC34 was a positive regulator of carotenoid biosynthesis in oriental melon fruit.

### 2.6. CmNAC34 Directly Binds to the Promoter of CmLCYB and Activates Its Expression

To identify how CmNAC34 regulated carotenoid accumulation, a Y1H assay was conducted, and the results showed that the yeast cell as positive control transformed with (pAbAi-p53 + pGADT7-Rec-53), negative control transformed with (*CmLCYB*pro-pAbAi + empty vector pGADT7), and CmNAC34 transformed with (*CmLCYB*pro-pAbAi + pGADT7-CmNAC34) all could grow on SD selection medium without Leu (SD/-Leu). However, only the yeast cells transformed with positive control and CmNAC34 could grow on the SD selection medium with the presence of 100 ng·mL^−1^ aureobasidin A (SD/-Leu/AbA^100^), which indicates that CmNAC34 could bind to the promoter of *CmLCYB* in the yeast system (Figure 5a). Additionally, an EMSA assay was conducted to further identify the binding site in the *CmLCYB* promoter. Here, the result implied that the CmNAC34 was able to bind the CATGTG motif in the *CmLCYB* promoter (Figure 5b).

Furthermore, the GUS assay was performed to identify the regulation of CmNAC34 on the promoter of *CmLCYB*. The results showed that the promoter activity of *CmLCYB* in tobacco leaves co-infiltrated with 35S::CmNAC34 and CmLCYBpro-GUS was significantly promoted compared with the control, and the color of tobacco leaves was also darker than that of the control (Figure 5c). Meanwhile, the luciferase reporter assay was also conducted to further confirm this result. When the tobacco leaves were co-injected with pRI-CmLCYBpro-LUC and 35S::CmNAC34, the fluorescence signal was significantly higher than that of the control (Figure 5d). These results implied that the CmNAC34 could directly bind to the *CmLCYB* promoter and promote its expression.

## 3. Discussion

Carotenoid accumulation is one of the important quality trait formations, along with the ripening of oriental melon fruit. As reported in a previous study [34], β-carotene is the major carotenoid component in oriental melon, and its function and structure differ from other carotenoids. The major advantage of β-carotene is that it is considered the most optimal and suitable precursor of vitamin A which plays diverse roles in human health, such as immune function, normal growth and development, and vision. In addition, β-carotene functions as an antioxidant and singlet oxygen quencher, which is also attributed to human health [35]. In this study, the β-carotene content sharply accumulated at the fruit maturity stage (from 25 to 35 DAA), which is consistent with the changes in fruit color in oriental melons (Figure 1a,c). This result further verified the previous study [34] that the β-carotene content is related to the fruit color of oriental melon. Moreover, the accumulation of β-carotene is partially associated with the transcription of *CmLCYB* in oriental melon. Similarly, the major carotenoid, β-carotene, accumulation in kiwifruit (*Actinidia*
*deliciosa*) appeared to be under the control of transcription of *AdLCYB* [12]. In the durian (*Durio zibethinus*), its pulp mainly accumulated β-carotene and α-carotene, which is highly correlated to the expression level of key genes such as *LCYB* and *CYCB* [36].

The regulatory mechanism at the transcription level was the dominant way for alteration of carotenogenesis expression to affect carotenoid concentration during fruit development [4]. The promoter region of *CmLCYB* was analyzed in detail to further explore its regulatory mechanism. Firstly, the promoter activity was represented by GUS activity and histochemical staining, which revealed that the promoter activity was gradually enhanced with the increase in promoter length (Figure 3b,c). This phenomenon may be explained by the number of enhancer elements that were elevated with the extension of the promoter. Then, the promoter region was characterized through bioinformatic analysis, and the result showed that apart from basic elements (TATA-box and CAAT-box), the promoter contains several *cis*-elements bound by TFs, such as NAC, MYB, ERF, WRKY, and MADS (Table 1). According to previous reporters, these TFs were all involved in carotenoid metabolism. For example, carotenoid biosynthesis was modulated by CpNAC1, activating the expression of *CpPDS2/4* during the papaya (*Carica papaya*) fruit ripening [30]. In kiwifruit (*Actinidia deliciosa*), AdMYB7 bound to the promoter of *AdLCYB* and promoted its transcription to affect carotenoid accumulation [37]. ERF transcription factor, CsERF06, induced by ethylene, activated carotenoid metabolic pathway genes to manipulate the carotenoid accumulation in citrus (*Citrus sinensis* L. Osbeck) [38]. Several SlWRKY TFs interact with and activate the promoter of *SlPSY1* and *SlPDS* to regulate carotenoid biosynthesis in tomato [21]. In citrus (*Citrus sinensis*), CsMADS5 and CsMADS6 independently and cooperatively regulate *LCYb1* and other carotenogenic genes to modify the carotenoid metabolism [39,40]. Additionally, MaSPL16 directly binds to the promoter of *MaLCYB*s (*MaLCYB1.1* and *MaLCYB1.2*) and stimulates their expression to alter the carotenoid biosynthesis in banana (*Musa acuminata*) [41]. Therefore, we paid close attention to these transcription factors and selected that they could regulate carotenoid metabolism using a series of assays in oriental melon.

Combining promoter analysis with yeast one-hybrid cDNA library screening, the CmNAC34 was identified, and multiple sequences alignment were performed with protein sequences. Here, the CmNAC34 shared 34.34%, 31.04%, 34.07%, and 33.77% identify with SlNAC1 and SlNAC4, SNAC9 (SlNAC19), SNAC4 (SlNAC48), and AtNAP, respectively, which indicates that CmNAC34 has the potential ability to regulating carotenoid metabolism. Therefore, the expression profile of *CmNAC34* was determined using RT-qPCR assay and was consistent with that of *CmLCYB* (Figure 3a), and CmNAC34 was located in the nucleus, functioned as transcriptional activators (Figure 3b,c). These results further imply that CmNAC34 served as a candidate gene regulating carotenoid metabolism.

Because there are some difficulties in the genetic transformation system of oriental melon, we performed an *Agrobacterium tumefaciens*-mediated fruit infiltration assay to further verify that CmNAC34 affected *CmLCYB*-mediated carotenoid metabolism. The expression of CmNAC34 was gradually increased and then slightly decreased, peaking at 4.5 days after injection in CmNAC34-OE fruit, which corresponded to the changes in luciferase fluorescence intensity (Figure 4a). Additionally, the transcription level of *CmNAC34* was consistent with *CmLCYB* not only in CmNAC34-OE fruit but in CmNAC34-AN fruit, which implied the CmNAC34 could regulate the expression of *CmLCYB* (Figure 4b–e). Then, the β-carotene content was significantly higher and lower in CmNAC34-OE fruit and CmNAC34-AN fruit, respectively (Figure 4f,g). The proposed working module is shown in Figure 6. It is worth noting that the expression of *CmNAC34* and *CmLCYB* or β-carotene content only showed a significant difference in individual time of transiently transgenic fruit, which may be explained by the reason that *Agrobacterium tumefaciens*-mediated fruit infiltration assay was susceptible to environmental factors, such as temperature and bacterial activity. However, from the view of the whole trance of expression profile and β-carotene content, we could conclude that the CmNAC34 manipulated *CmLCYB*-mediated carotenoid metabolism in oriental melon. Of course, the stable transgenic system of CmNAC34 needs to be conducted in future studies.

Finally, the Y1H, EMSA, GUS analysis, and luciferase reporter assay were conducted to further prove the effect on the promoter of *CmLCYB*, and as expected, the CmNAC34 stimulated the transcription of *CmLCYB* by directly binding to its promoter. In addition, CpEIN3a interacted with CpNAC2 and cooperatively regulated carotenoid biosynthesis-related genes to control carotenoid production in papaya (*Carica papaya*) [31]. However, the promoter of *CmLCYB* also contained other regulatory *cis*-elements, and whether other transcription factors individually or as a partner of CmNAC34 cooperatively modulate carotenoid metabolism in oriental melon, which is one of the priorities of our next study.

## 4. Materials and Methods

### 4.1. Plant Materials

The orange-flesh oriental melon cultivar ‘HDB’ and white-flesh oriental melon cultivars ‘YMR’ and ‘XSM’ (*Cucumis melo* var. *makuwa* Makino, 2n = 2x = 24) were used in the study and planted in a greenhouse at Shenyang Agricultural University, Shenyang, China, from March through July. Female flowers were treated with 2.5 mg L^−1^ N-(2-Chloro-4-pyridyl)-N′-phenylurea (CPPU) to promote fruit set and mark the date of pollination. Three fruit were allowed to develop on identical sites on each plant, and the fruit was sampled at 5, 10, 15, 20, 25, 30, and 35 DAA. The fruit was immediately washed after harvest, and the peels and stalks were removed. The equator pulp was immediately collected, frozen in liquid nitrogen, and stored at −80 °C. Roots, stems, leaves, flowers, and tendrils were sampled at the five-leaf stage.

### 4.2. β-Carotene Content Detection by HPLC and Color Measurement

The β-carotene was extracted and quantified as described previously with some modifications [42]. Briefly, 3 g of fruit flesh was ground to a fine juice with 5 mL ethanol with 0.1% BHT in an ice bath. Subsequently, the fine juice was transferred to a sintered filter funnel, and the oriental melon juice was washed with absolute ethanol (0.1%BHT), methyl alcohol (0.1% BHT), and 2% dichloromethane-petroleum ether (0.1% BHT), respectively, until the extracts were colorless. Then, the β-carotene solution was transferred to a brown separating funnel, the water phase was dropout, and the organic phase was transferred into 50 mL brown centrifuge tubes. These solutions were dried using a concentration vacuum centrifuge and dissolved with 5 mL dichloromethane. For HPLC detection, the samples were filtered using 0.22 μm nylon membrane and measured using HPLC (Waters 2695-series HPLC, Waters Corp., Milford, MA, USA) equipped with a UV-detector (waters e2489) and C18 column (250 × 4.6 mm, 5 μm, Milford, MA, USA). The mobile phase was comprised of methanol:acetonitrile:dichloromethane (2:5:3, *v*/*v*/*v*), column temperature of 25 °C, flow rate of 0.8 μL, and injection volume of 10 μL. β-carotene was identified through the retention time consistent with β-carotene standard detecting at 502 nm and calculated using its standard curves.

For flesh color measurement, the chromameter (CR-400, Konica Minolta, Tokyo, Japan) was used to determine the flesh color of oriental melon at different developmental stages. Three random sites of each fruit were selected, and at least three fruits were tested. Then, the L, a*, and b* values were generated automatically, and their mean values quantified the flesh color of the oriental melon.

### 4.3. Promoter Isolation, Bioinformatics Analysis, and Transient Assay of Promoter Activity

The 5′-upstream promoter sequence (1474bp) of *CmLCYB* was obtained using PCR amplification, and the PCR product was subcloned into a 19-T vector (Takara, Dalian, China). After sequencing, the true promoter of *CmLCYB* was selected and submitted to the PlantCARE (http://bioinformatics.psb.ugent.be/webtools/plantcare/html/) (accessed on 27 April 2022) and new PLACE (https://www.dna.affrc.go.jp/PLACE/?action=newplace) (accessed on 2 May 2022) database to predict the diverse cis-elements. For promoter activity analysis, we deleted different lengths from the 5′ region to produce three promoter fragments. These promoter fragments were fused to GUS by replacing the CaMV35S promoter in the PBI101-GUS vector, respectively, while the PBI101 empty vector was used as a negative control. Subsequently, these fusion plasmids were transferred into Agrobacterium tumefaciens strain EHA105 and cultured on a YEP plate at 28 °C for 48 h. The healthy tobacco (Nicotiana benthamiana) leaves were used for injection and grown in a chamber for three days to detect GUS activity. The GUS activity and histochemical staining in transiently expressed tobacco leaves were detected as described previously [43]. The infiltration assay was conducted with three biological replicates.

### 4.4. Genes Expressions Profile Analysis

Total RNA extraction from oriental melon fruit was performed using an RNA Isolation Kit (Cat #CW0581; CWbio Co., Ltd., Beijing, China) according to the manufacturer’s instructions. Subsequently, the first-strand cDNA synthesis was conducted using the PrimeScript™ RT reagent kit with a genomic DNA Eraser (Takara, Dalian, China) according to the manufacturer’s instructions. The cDNA was used as a template for the qRT-PCR assay, and the qRT-PCR was performed on qTOWER3G (Analttik Jena AG, Jena, Germany), followed by the SuperReal PreMix Plus (SYBR Green) (Takara, Dalian, China) manufacturer’s protocol. The relative expression level was analyzed from three biological and technical replicates using the 2^−ΔΔCt^ method, and 18 s rRNA was used as the internal control.

### 4.5. Yeast One Hybrid cDNA Library Screening in Oriental Melon Fruit

The yeast one hybrid cDNA library screen was conducted as previously described [44]. Briefly, the promoter of *CmLCYB* was subcloned into the pAbAi vector to generate a bait vector. Subsequently, the bait vector was linearized using BstBI, and the linearized vector was transformed into Y1H gold according to the Matchmaker™ Gold Yeast One-Hybrid Library Screening System kit (Cat. no. 630491, Clontech, Mountain View, CA, USA). The transformed yeast cell with linearized vectors was screened on SD/Ura medium with different concentrations of aureobasidin (AbA), and the yeast cell with the correct fragment was verified using PCR for further analysis. Finally, the yeast one hybrid cDNA library was transformed into the yeast cell with the correct fragment and cultured on SD/Leu (AbA) at 30 °C for 3–5 d. The healthy yeast clone was sequenced and analyzed.

### 4.6. Subcellular Localization and Transcriptional Activation

The CDS of CmNAC34 missing stop codon was amplified and subcloned into the pCAMBIA1300 vector with a GFP to generate a 35S::CmNAC34-GFP fusion construct, and an empty pCAMBIA1300 vector was used as the positive control. These plasmids were introduced into tobacco leaves through Agrobacterium tumefaciens infiltration. Three days after the injection, tobacco leaves were collected, and the GFP fluorescence was observed using a laser confocal fluorescence microscope (TCS SP8-SE; 158 Leica, Wetzlar, Germany) after being stained by the nucleus marker DAPI.

For transcriptional activation assay, the CDS of *CmNAC34* was subcloned into the pGBKT7 vector (Clontech, Mountain View, CA, USA) to produce pGBKT7-CmNAC34 fusion plasmid. The empty pGBKT7 vector and pGBKT7-53 + pGADT7-RecT were used as the negative and positive control, respectively. These plasmids were separately transformed into Y2H Gold yeast cells according to the manufacturer’s instructions (Clontech, Mountain View, CA, USA). The transformed yeast cell was cultured on amino acid-deficient medium SD/-Trp for 3–5 days. Then, several positive colonies were dotted on SD/-Trp-His-Ade with X-α-gal and incubated for 3–5 days at 30 °C to evaluate the transcriptional activity.

### 4.7. Transient Expression of CmNAC34 in Oriental Melon Fruit

Due to some difficulties existing in the genetic transformation system of oriental melon, the *Agrobacterium* tumefaciens-mediated transient expression assay was conducted in oriental melon to further verify the function of CmNAC34. In order to improve the expression of *CmNAC34* in oriental melon, the CDS region was inserted into the pCAMBIA3301-LUC vector and under the control of the 35S promoter to generate the 35S::CmNAC34-LUC construct (CmNAC34-OE). The empty pCAMBIA3301-LUC (LUC-0) was used as the control. To silence the expression of *CmNAC34* in oriental melon, the CDS sequence was cloned into the PRI101 vector in the reverse direction to yield the antisense CmNAC34 construct (CmNAC34-AN), and the empty PRI101 vector (PRI101-0) was the control. These plasmids were transformed into *Agrobacterium tumefaciens* strain EHA105, and the fruit infiltration assay was conducted as described previously [45]. In brief, 300 μL of infiltration buffer was injected into ‘HDB’ fruit on the plant with a 1 mL sterile syringe at 25 DAA, and each fruit was injected at 6 sites. The infiltrated fruit was harvested at 0, 1.5, 2.5, 4.5, 6.5, and 8.5 days after injection, and the fruit flesh around the injection sites was sampled for further study. The luciferase fluorescence was detected, followed by the previous reporter [45]. One fruit was regarded as a biological replicate, and at least three replicates were used.

### 4.8. Y1H Analysis

The promoter of *CmLCYB* was amplified and inserted into the pAbAi vector to form a bait construct, pAbAi-CmLCYBpro. The CDS of *CmNAC34* was fused to the pGADT7 vector to produce a prey construct, pGADT7-CmNAC34. For the Y1H assay, the bait plasmid was linearized and then integrated into the Y1H Gold yeast genome to generate bait strains. The prey plasmids were transformed into generated bait strains, and the yeast cell was plated on amino acid-deficient medium SD/-Leu with or without antibiotics for 3–5 days at 30 °C. Y1H assay was performed using the Matchmaker^TM^ Gold Yeast One-Hybrid Library Screening System Kit (Cat. No. 630491; Clontech, Mountain View, CA, USA).

### 4.9. EMSA

The EMSA was performed as previously described [46]. The CDS sequence of *CmNAC34* was cloned into the pGEX-6p vector (with GST-tag) and transformed into Rosetta (DE3) Escherichia coli cell to express the CmNAC34 protein, and the recombinant protein was purified using ProteinIso^®^ GST Resin kit (TransGen) according to the manufacturer’s protocol. The 5′ biotin end-labeled probes were synthesized by Sangon Biotech. The EMSA was performed using the EMSA kit (GS009; Beyotime, Shanghai, China) according to the manufacturer’s protocol with probes.

### 4.10. GUS Analysis

The promoter fragment of *CmLCYB* was cloned into the PBI101 vector by replacing the 35S promoter to yield the CmLCYBpro::GUS reporter vector. The CDS of *CmNAC34* was inserted downstream of 35S in the PRI101 vector to produce 35S::CmNAC34 as an effector, and the empty PRI101 vector was used as the negative control. Agrobacterium tumefaciens stain EHA105 was transformed with all resulting vectors and incubated at 28 °C until the OD value reached 1.0. The tobacco leaves were infiltrated with Agrobacterium tumefaciens stains separately harboring reporter and effector vector at a ratio of 1:1. Three days after injection, the tobacco leaves were collected, and the GUS activity was determined as previously described [43].

### 4.11. LUC Assay

The CDS of *CmNAC34* was amplified and ligated into the PRI101 vector to yield an effector construct, 35S::CmNAC34. The promoter region of *CmLCYB* was cloned into the pRI-mini35S-LUC vector to produce a reporter construct, and the empty pRI-mini35S-LUC was used as a control. All resulting constructs were transformed into Agrobacterium tumefaciens stain EHA105. Tobacco leaves were used for co-infiltration and treated with 1 mM D-Luciferin Potassium Salt (Thermo Fisher Scientific, Waltham, MA, USA) after three days. The luciferase fluorescence signal was detected using the NightShade LB 985 In Vivo Plant Imaging System (Berthold, Bad Wildbad, German), and the fluorescence image was taken using the IngiGo program.

## 5. Conclusions

We identified that the β-carotene massively accumulated during the oriental melon fruit development and ripening. CmLCYB was the critical enzyme that directly synthesized β-carotene, and it was selected as one of the key genes for further study. The bioinformatic analysis of *CmLCYB* promoter and yeast one-hybrid screening with *CmLCYB* promoter were conducted, and the CmNAC34 was identified as the candidate TF. We found that the expression profile of *CmNAC34* was consistent with that of *CmLCYB* during the fruit ripening and its function as a transcription activator. The putative hypothesis was proposed that the CmNAC34 may regulate β-carotene accumulation by promoting the transcription of *CmLCYB*. Finally, our results confirmed that CmNAC34 could activate the expression of *CmLCYB* by directly binding to its promoter, thereby regulating *CmLCYB*-mediated β-carotene accumulation. In summary, CmNAC34 could modulate β-carotene accumulation by improving the *CmLCYB* expression in oriental melon.

## Figures and Tables

**Figure 1 ijms-23-09805-f001:**
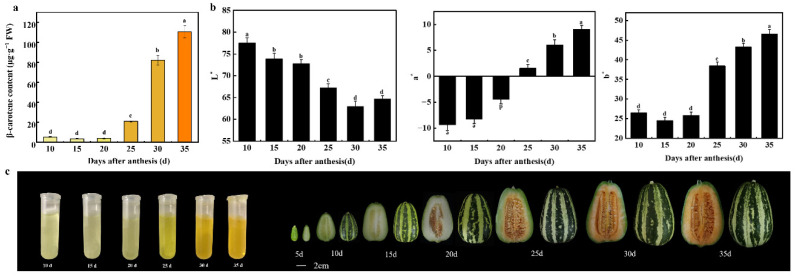
Characterization of fruit flesh in oriental melon. (**a**) β-carotene content of fruit flesh in ‘HDB’ during different development stages. Values represent the mean ± SE of three replicates. Different letters represent a statistically significant difference at the different days after anthesis (*p* < 0.05). (**b**) L*, a*, and b* were measured during the fruit development and ripening. Values represent the mean ± SE of three replicates. (**c**) Changes in fruit flesh of ‘HBD’ during different developmental stages.

**Figure 2 ijms-23-09805-f002:**
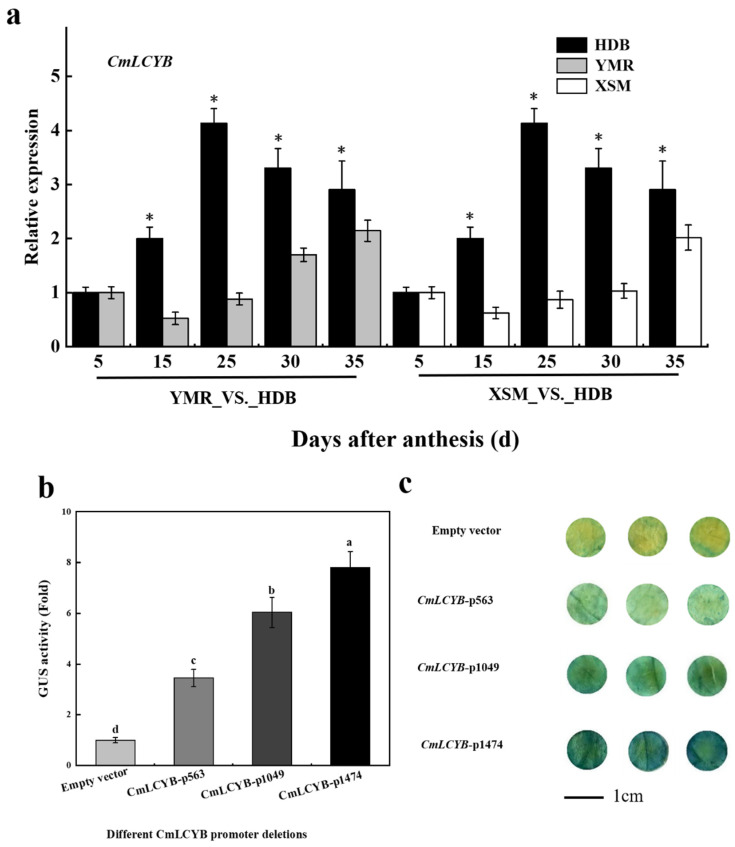
*CmLCYB* expression profile and its promoter activity. (**a**) The expression of *CmLCYB* in different flesh-color oriental melons during the different development stages. Values represent the mean ± SE of three replicates. Asterisks indicate significant differences determined by the independent sample *t*-test (* *p* < 0.05). (**b**) GUS activity of different promoter fragments. Values represent the mean ± SE of three replicates. Different letters represent a statistically significant difference at the different days after anthesis (* *p* < 0.05). (**c**) GUS histochemical staining of different promoter fragments.

**Figure 3 ijms-23-09805-f003:**
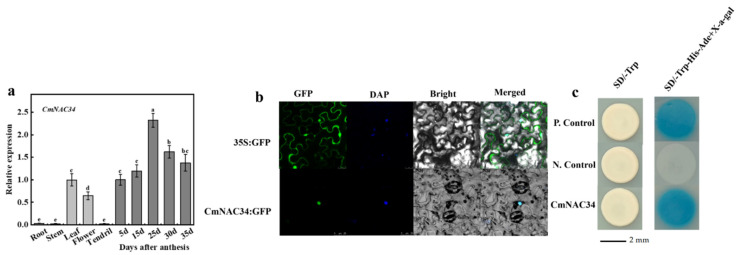
Expression profile and functional analysis of CmNAC34. (**a**) Temporal and spatial expression profile of *CmNAC34* in ‘HDB’. Values represent the mean ± SE of three replicates. Different letters represent a statistically significant difference (*p* < 0.05). (**b**) Subcellular localization of CmNAC34, GFP is green fluorescent protein (green); DAPI is nucleus dye (blue); bright filed, white light; merged is the combination of GFP signal with DAPI (cyan). (**c**) Transcription activity of CmNAC34 in the yeast system. CmNAC34 was inserted into the pGBKT7 vector and transformed into yeast cell Y2H Gold. The yeast cell was cultured in amino acid-deficient medium SD/-Trp or SD/-Trp-His-Ade with X-α-gal. P. Control, positive control (pGBKT7-53 + pGADT7-RecT), N.Control, negative control (empty pGBKT7 vector), CmNAC34, pGBKT7-CmNAC34 vector.

**Figure 4 ijms-23-09805-f004:**
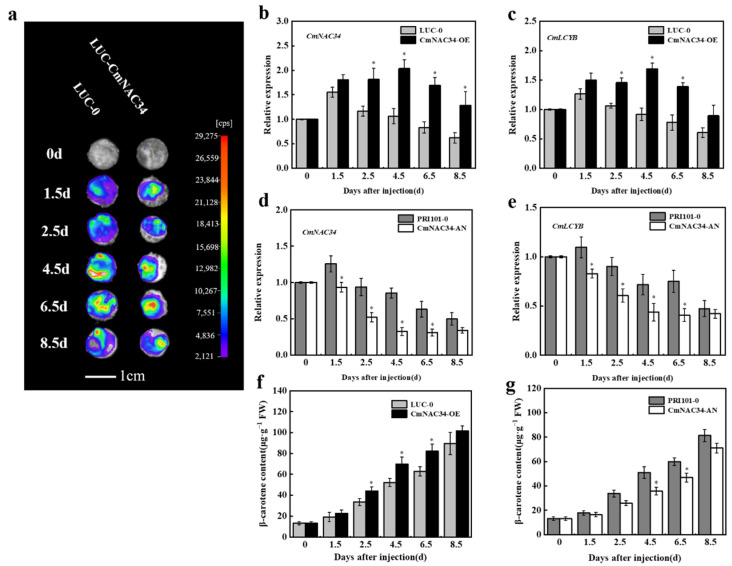
*CmNAC34* was transiently expressed in oriental melon and β-carotene concentration in transiently transgenic fruit. (**a**) The luciferase intensity was detected in LUC-0 and LUC-CmNAC34 fruit at different days after infiltration. (**b**,**c**) The relative expression of *CmNAC34* and *CmLCYB* in oriental melon fruit overexpressed CmNAC34 and fruit infiltrated with LUC-0. (**d**,**e**) The relative expression of *CmNAC34* and *CmLCYB* in oriental melon fruit silenced CmNAC34 and fruit infiltrated with PRI101. (**f**) The β-carotene concentration of transient overexpression of CmNAC34 and LUC-0 fruit were measured using HPLC. (**g**) The β-carotene concentration of transient silence of CmNAC34 and PRI101fruit were measured using HPLC. The data were analyzed from three biological replicates, and the error bar represents the SE. Asterisks indicate a significant difference compared to control, as determined by the independent sample *t*-test (* *p* < 0.05).

**Figure 5 ijms-23-09805-f005:**
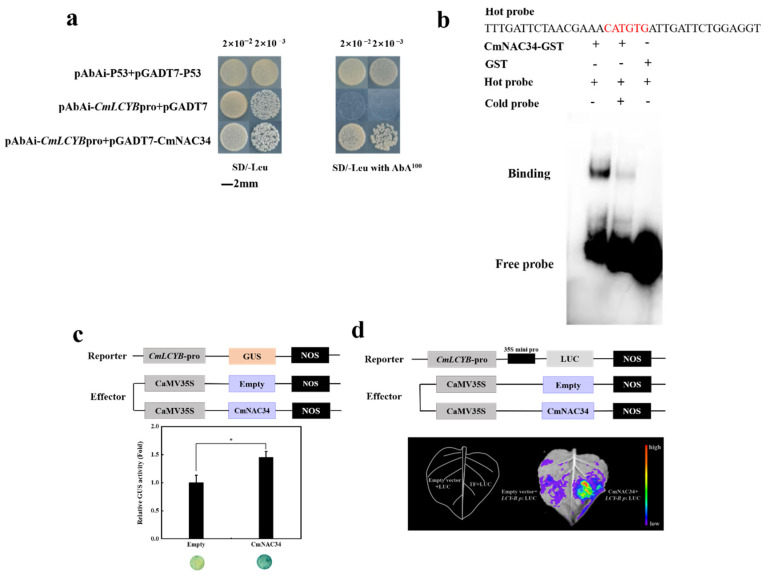
CmNAC34 activates the expression of *CmLCYB* by interaction with its promoter. (**a**) Y1H assay showed that CmNAC34 binds to the promoter of *CmLCYB* in the yeast system. SD/-Leu, SD medium deficient Leu; SD/Leu with AbA^100^, SD medium deficient Leu supplemented with 100 ng·mL^−1^ of AbA. The transformed yeast cells were dripped at 2 × 10^−2^ and 2 × 10^−3^ on the selective medium, respectively. (**b**) EMSA shows that the CmNAC34 binds to the CATGTG motif in the *CmLCYB* promoter. (**c**) Schematic diagram of reporter and effector constructs in GUS assay. The reporter and effector were co-infiltrated into tobacco leaves to analyze the regulation of GUS activity. (**d**) Schematic diagram of reporter and effector constructs in a luciferase reporter assay. The reporter and effector were co-infiltrated into tobacco leaves, and this assay showed CmNAC34 positively regulated *CmLCYB* expression. The colors represent the gene expression abundance. Three independent injected tobacco leaves were performed, and the error bar represents the SE. Asterisks indicate a significant difference compared to control, as determined by the independent sample *t*-test (* *p* < 0.05).

**Figure 6 ijms-23-09805-f006:**
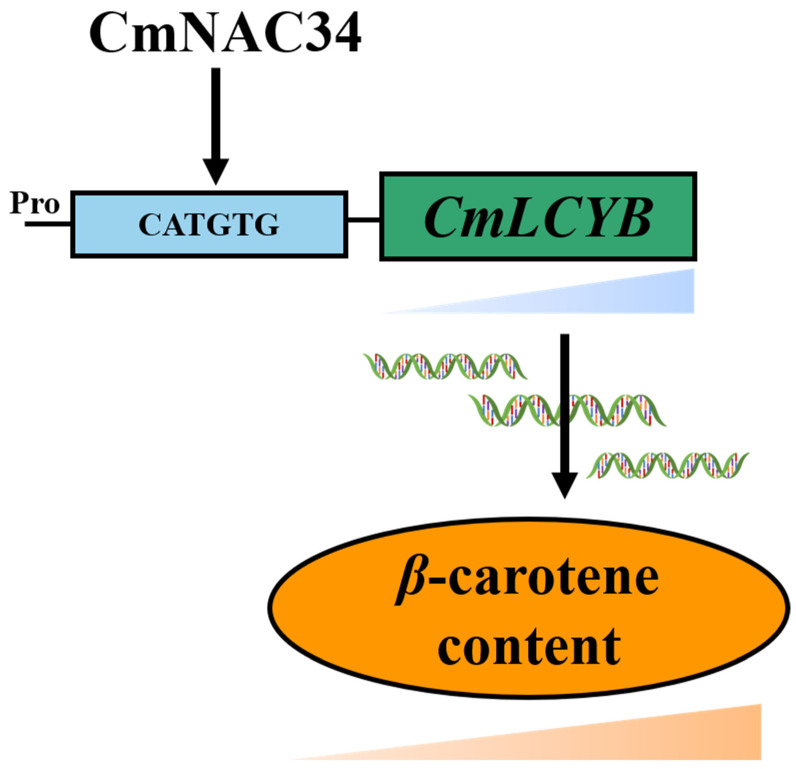
Proposed model of CmNAC34 function in *CmLCYB* regulating β-carotene accumulation during the oriental melon fruit ripening. The CmNAC34 was involved in carotenoid metabolism by directly binding to the promoter of *CmLCYB* and activating its transcription.

**Table 1 ijms-23-09805-t001:** The putative regulatory elements were identified in the *CmLCYB* promoter region.

*Cis*-Element	Sequence	Function of Site
ABRE	CGTACGTG	ABA-responsive element
CGTCA-motif	CGTCA	MeJA-responsive element
NAC-motif	ACATGTG	NAC binding site
ARE	AAACCA	Anaerobic-responsive element
CAAT-box	CAAT	Core promoter element
TATA-box	TATATA	Common enhancer element
GARE-motif	TCTGTTG	Gibberellin-responsive element
GT1-motif	GGTTAA	Light-responsive element
MBS	CAACTG	Drought-responsive element
MADS-box	CCTAAA	MADS binding site
MYB	TAACCA	MYB binding site
MYC	CAATTG	MYC binding site
TCT-motif	TCTTAC	Light-responsive element
TGA-element	AACGAC	Auxin-responsive element
ERE	ATTTTAAA	Ethylene-responsive element
W-box	TTGACC	WRKY binding site

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
