# Peer review of "Transcription Factor CmNAC34 Regulated *CmLCYB*-Mediated β-Carotene Accumulation during Oriental Melon Fruit Ripening"

_ijms, 2022, doi:10.3390/ijms23179805_

Round 1

Reviewer 1 Report

Dear author(s),

The following points should clearly be explained and addressed for readers:

Abstract

1.    Line (L) 12, scientific name of melon should be written in the first mentioned place.

2.    L23, GUS could be written/explained as LUC before abbreviation

Keywords

3.    L27, scientific name of melon could be written.

Introduction

4.    L48, change However to Moreover,

5.    L52-60, scientific name could be written with author name. Also, check whole text.

6.    L85-91, scientific name?

7.    L92-103, aim of the study should be written clearly.

Results

8.    L140-142, In Figure 2, please insist on P/p and they should be italicized. Check whole text.

9.    Please check whole abbreviations. Abbreviations should be used in the first mentioned place. Abbreviations should be removed M&M to Results because Results were given first according to guide for authors.

10. Please insist on italic gene symbols in whole text and check for this suggestion.

11. L202, subtitles should be consistent. Please write lowercase/capital letters or lowercase/small letters for each word.

12. L213, The, Its etc. should be written with small letters. Please check whole text for this.

Discussion

13. Discussion was the best

Conclusions

14. L490, delete in the study because readers know “this study”

15. L496, I suggest possible or putative hypothesis so you can write putative or possible hypothesis

16. but in L347, please write the main findings and a conclusion sentence for readers.

Reviewer 2 Report

Thank you for inviting me to review the ms titled Transcription Factor CmNAC34 Regulated CmLCYB-mediated 2 β-Carotene Accumulation during Oriental Melon Fruit Ripen- 3 ing

The authors try to understand the regulatory mechanisms underlying carotenoid metabolism in oriental melon. They identified CmNAC34 by yeast one-hybrid screening, which regulated carotenoid accumulation by activating the expression of CmLCYB during the fruit ripening.

The results are interesting. They only need to do some minor corrections as follow:

Do a better literature review and address more new relevant papers. For example you can address to the following papers:

Mohammad Reza Raji, Mahmoud Lotfi, Masoud Tohidfar, Hossein Ramshini, Navazollah Sahebani, Mostafa Aalifar, Mahnaz Baratian, Francesco Mercati, Roberto De Michele, Francesco Carimi, Multiple fungal diseases resistance induction in Cucumis melo through co-transformation of different pathogenesis related (PR) protein genes, Scientia Horticulturae, Volume 297, 2022, 110924

Sheikh Beig Goharrizi MA, Dejahang A, Tohidfar M, Izadi Darbandi A, Carillo N, Hajirezaei MR and Vahdati K (2016) Agrobacterium mediated transformation of somatic embryos of Persian walnut using fld gene for osmotic stress tolerance. Journal of Agricultural Science and Technology. 18: 423-435.

Nazari M, Tohidfar M, Ramshini H, Vahdati K (2022) Molecular and morphological evaluation of transgenic Persian walnut plants harboring Fld gene under osmotic stress condition. Molecular Biology Reports 49 (1), 433-441.

Rezaei Qusheh Bolagh F, Solouki A, Tohidfar M, Zare Mehrjerdi M, Izadi-Darbandi A, Vahdati K (2020) Agrobacterium-mediated transformation of Persian walnut using BADH gene for salt and drought tolerance, The Journal of Horticultural Science and Biotechnology, 95(4): 1-10.

I recommend to edit the paper by a native English editor.
